# Classroom Behavior Detection Based on Improved YOLOv5 Algorithm Combining Multi-Scale Feature Fusion and Attention Mechanism

**Longyu Tang [1,†], Tao Xie [2,†], Yunong Yang [1,*] and Hong Wang [1]**

[1]  College of Computer and Information Science, Chongqing Normal University, Chongqing 401331, China; 2020210516092@stu.cqnu.edu.cn (L.T.); tonywanghong@126.com (H.W.)

[2]  Faculty of Education, Southwest University, Chongqing 400715, China; xietao@swu.edu.cn

\*  Correspondence: yangyunong@cqnu.edu.cn

†  These authors contributed equally to this work.

**Abstract:** The detection of students' behaviors in classroom can provide a guideline for assessing the effectiveness of classroom teaching. This study proposes a classroom behavior detection algorithm using an improved object detection model (i.e., YOLOv5). First, the feature pyramid structure (FPN+PAN) in the neck network of the original YOLOv5 model is combined with a weighted bidirectional feature pyramid network (BiFPN). They are subsequently processed with feature fusion of different scales of the object to mine the fine-grained features of different behaviors. Second, a spatial and channel convolutional attention mechanism (CBAM) is added between the neck network and the prediction network to make the model focus on the object information to improve the detection accuracy. Finally, the original non-maximum suppression is improved using the distance-based intersection ratio (DIoU) to improve the discrimination of occluded objects. A series of experiments were conducted on our new established dataset which includes four types of behaviors: listening, looking down, lying down, and standing. The results demonstrated that the algorithm proposed in this study can accurately detect various student behaviors, and the accuracy was higher than that of the YOLOv5 model. By comparing the effects of student behavior detection in different scenarios, the improved algorithm had an average accuracy of 89.8% and a recall of 90.4%, both of which were better than the compared detection algorithms.

**Keywords:** classroom behavior detection; attention mechanism; pyramid network; non-maximal suppression

## 1. Introduction

With the continuous development of artificial intelligence, intelligent education has become a popular topic in recent years [1]. Students are the main body of classroom learning activities. The introduction of artificial intelligence into classroom teaching activities and the use of deep learning to identify students' behaviors in the classroom will help to understand the students' learning status and improve classroom teaching efficiency. For example, if a student is attracted by interesting materials, he will show signs of listening, standing up, or engaging in Q&A with the teacher. If a student is bored in class, he will put his head down, get distracted, and even sleep on his desk. Therefore, the detection of students' behavior in the classroom is of great significance. However, large numbers of students sitting in classroom scenarios can lead to severe object occlusion, which poses a significant challenge to behavior detection. In traditional methods, students' behavior in class is recorded manually, which not only results in the lack of behavior record, but also consumes a lot of human resources. The introduction of the Flanders' interactive system marked the beginning of the modern quantitative analysis classroom [2]. Various optimization analysis systems have been devised, but most of them still rely on teachers'

judgment or observations after class. Therefore, traditional teaching evaluation lacks automatic classroom behavior analytical tools.

In 2006, scholars proposed the concept of deep learning, suggesting the use of computers to perform high-latitude matrix operations to describe the properties and characteristics of objects at a deeper level [3]. With the development of computer performance, image feature extraction methods based on convolutional neural networks (CNN) such as VGGNet [4], GoogleNet [5], and ResNet [6] were proposed to increase the accuracy of image recognition and feature extraction. The authors of [7] used ConvNet for feature extraction from complex EEG signals and achieved high accuracy, demonstrating the superiority of deep learning-based convolutional neural networks in the field of image processing. In the library of behavior recognition based on deep learning, the two-stream convolutional neural network input RGB images and optical flow images into the network separately to extract spatial and temporal features [8,9]. The 3D convolution algorithm adds temporal dimension to the 2D CNN [10], and learns both spatial and temporal information. The authors of [11] proposed ResNet based on a 3D network to further improve the accuracy of behavior recognition. On this basis, a pseudo 3D network (P3D) that simulates a 3D neural network in 2D is proposed in [12]. However, these methods require a complete analysis of a video, which is computationally expensive and cannot mark the position of each behavior. Therefore, it is not appropriate to conduct behavior detection in multi-person classroom scenarios. As shown in Figure 1, the dataset used in this paper was developed in an ordinary classroom environment. The goal of this study was to identify multiple students' posture states in the classroom, such as listening, looking down, lying down, and standing. These classroom behaviors are difficult to recognize using existing behavior recognition algorithms because of occlusions among students. The skeletal keypoint-based detection method identifies behavior by learning changes in a person's joint points [13], but it is difficult to pinpoint each person's joint points in a complex classroom scene.

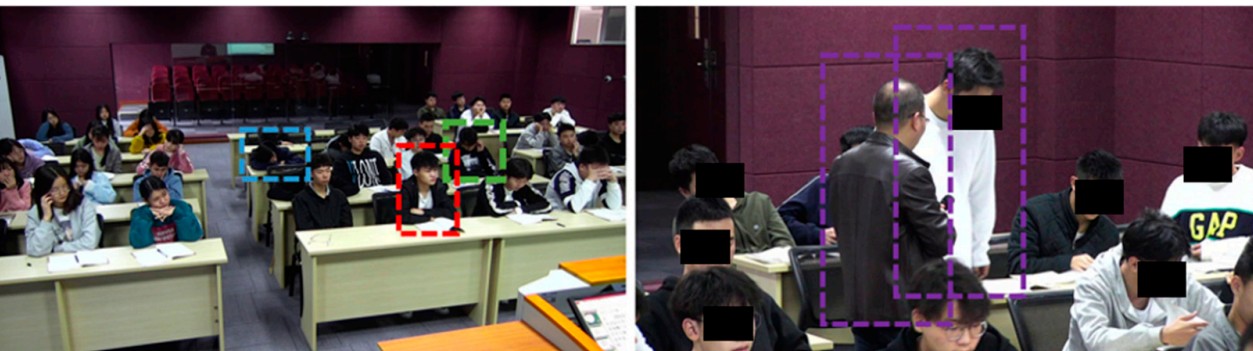

**Figure 1.** Some representative images in the classroom scenes. The red dashed boxes denote 'listening' samples, the green dashed boxes denote 'looking down' samples, the blue dashed boxes denote 'lying down' samples, and the purple dashed boxes denote 'standing' samples.

Classroom behavior detection is a non-trivial task because there are typically a number of students in the same space, and the front and back rows of students do not have the same proportion of pixels in the image which causes severe occlusion issues. In order to detect classroom behaviors more accurately, the detection of the fine-grained features regarding these occlusions is one of the most important tasks. Moreover, the students' behavior is less variable due to their relatively fixed positions during the class. The object categories are actually static postures or appearances, e.g., listening, looking down, and standing, which do not require multiple frames of temporal information. Therefore, this study does not consider the temporal characteristics of the behavior in order to maintain the accuracy and speed of recognition.

The authors of [14] used the object detection approach and proposed a Faster R-CNN algorithm combining ROI pooling and local hold learning to classify classroom behavioral images. They transformed the problem of classroom behavior detection into a fine-grained classification problem of behavioral images. However, the inference speed of the two-stage detection framework is slow, which cannot be used in practical applications. To address this problem, this study follows this line of research, using the object detection algorithm to locate the targets of students in the classroom and perform behavioral classification.

We propose a superior solution for classroom behavior detection by combining both detection accuracy and speed. In order to serve real classroom scenarios and improve detection efficiency, we focus on single-stage detection algorithms proposed in [15] and claim that this kind of algorithm is mainly affected by two factors: the learning of multi-scale fine-grained features and non-maximum suppressed post-processing methods. The mining of multi-scale features is crucial for detecting the fine-grained features of dense objects, and non-maximum suppression post-processing methods are key to solving the occlusion problem and improving the detection rate.

In summary, this study has the following contributions:

1.  We propose an end-to-end single-stage classroom behavior detection algorithm using a bidirectional weighted feature pyramid network to enhance the learning of multi-scale features.
2.  The intersection ratio strategy based on the centroid distance with a penalty term is used to improve the post-processing problem of non-maximum suppression, so as to address the problem of repeated detection and false detection in severely occluded scenes.
3.  A series of experiments are conducted to show that the proposed algorithm outperforms the current mainstream detection algorithms in classroom behavior detection with respect to the detection accuracy.

## 2. Related Work

### 2.1. Object Detection

The basic task of object detection is to determine the object categories to be detected in the image and give the corresponding confidence. In addition, the rectangular border is used to determine the location and size of the object. It is also an active and rapidly growing area of computer vision. Until the development of deep learning methods, the field of object detection developed slowly. In the ImageNet classification task in 2012, the application of convolutional neural networks greatly improved the effect of image classification tasks. Modern object detection methods are based on CNN, which can be divided into two-stage methods and single-stage methods. The authors of [16] proposed the first two-stage object detection algorithm called R-CNN. First, the region proposal network was used to extract candidate frames from the image, and then the object of the candidate frame was re-corrected to obtain the detection results. The authors of [17] proposed SPP-Net in 2014, using a spatial pyramid pooling layer to solve the problem of image distortion caused by image scaling. Subsequently, Fast R-CNN [18], Faster R-CNN [19], and Mask R-CNN [20] were also proposed. The two-stage object detection algorithm has the problem of slow detection speed. In order to speed up the inference, the single-stage object detection algorithm has been proposed. The YOLO algorithm directly exploits the entire image by using a single neural network without the need for a region proposal network. On this basis, SSD [21] and RetinaNet [22] algorithms were proposed successively, using contextual information or feature pyramid network to solve the problem of object multi-scale.

Compared with the two-stage algorithm, The single-stage object detection algorithm has faster detection speed, but the detection of small objects is not ideal, especially in the case of severe occlusion. In order to solve these problems, a multi-scale feature fusion strategy based on the YOLOv5 algorithm is adopted, which is combined with attention network and improves the non-maximum suppression post-processing.

## 2.2. Behavior Detection in Classroom Scenarios

In a related study of behavior detection in classroom scenes, the authors of [23] used a matrix of pressure sensors mounted on seats and the backs of chairs to collect students' various postures. The extracted posture features are fed into a feedforward neural network for training, and nine kinds of postures are classified with high precision. The authors of [24] used Hexiware to collect students' heartbeat data and categorize them by the K-Means algorithm as a measure of students' learning status in the classroom. The authors of [25] proposed a student body gesture recognition method based on the Fisher Broad learning system, and defined seven learning behaviors, which achieved good results on the self-built dataset. The authors of [26] used OpenPose framework to collect students' skeleton information, build a neural network to classify the extracted skeletal data by normalizing joint position, joint distance, and skeletal angle, and propose a student behavior recognition system based on human skeleton estimation and person detection. The authors of [27] built a deep convolutional neural network to identify head poses, used cascaded facial feature point positioning to extract facial expression key points, and distinguished students' classroom behaviors by combining head poses and facial expressions. The authors of [28] proposed a face tracking algorithm based on area of interest to detect students' standing behavior in the classroom, and also developed an algorithm based on skin color detection to recognize students' hand behavior. The authors of [29] introduced a cascaded RFB module in the YOLOv3 algorithm, which improves the feature extraction capability of the original network and realizes the goal of identifying small and medium-sized target students in the classroom. At the same time, the SE attention mechanism was introduced to express feature information in a finer-grained manner.

Through the above research and analysis, we can find that there are still some shortcomings in classroom behavior detection. (1) Although the behavior detection method using hardware has a high recognition rate, it needs to use hardware for multi-modal data collection, which is difficult to be applied to real classroom scenarios and is not suitable for large-scale popularization. (2) When using deep learning methods in the classroom, some algorithms cannot detect multiple objects in the same image frame at the same time and are not suitable for classroom scenarios with multiple students; or, although multiple people can be detected in the same frame, the algorithm lacks real-time performance. There is still a lack of a real-time classroom behavior detection methods. (3) The lack of classroom behavior datasets: there are currently no public large-scale classroom scene datasets, which makes it difficult for researchers to train in deep learning.

To address the above points, this paper marks the datasets of classroom behavior detection on real class scenarios and optimizes the behavior detection model according to the difficulties in class scenarios, and the optimized algorithm was tested to achieve good results.

## 3. Methods

Our purpose is to accurately detect a variety of student behaviors in the classroom environment in real time, so both detection accuracy and inference speed are important. At present, there is a problem of slow inference in Fast R-CNN and other two-stage object detection algorithms, so we consider using the YOLOv5 algorithm with good real-time performance.

### 3.1. YOLOv5 Framework

The YOLO series algorithm is a typical single-stage regression detection algorithm, which can accomplish both object localization and object classification at the same time. The YOLOv5 combines many of the advantages of previous versions [30,31], and strikes a good balance between accuracy and speed of detection. The YOLOv5 network structure is mainly composed of the input, the backbone network, the neck network, and the prediction head.

Input: YOLOv5 first resizes the incoming images of different sizes to 640 × 640 size, and completes the tasks of mosaic data enhancement, adaptive anchor frame calculation, and adaptive image scaling at the input side, and then sends them to the backbone network.

Backbone: The backbone network is used for feature extraction and consists mainly of Focus module, CSP module [32], and CBL module. The most critical part of the Focus structure is the slicing operation, as shown in Figure 2, where the original input size of 640 × 640 × 3 is sliced into a 320 × 320 × 12 feature map, and then undergoes a convolution operation with 32 convolution kernels to finally become a 320 × 320 × 32 feature map.

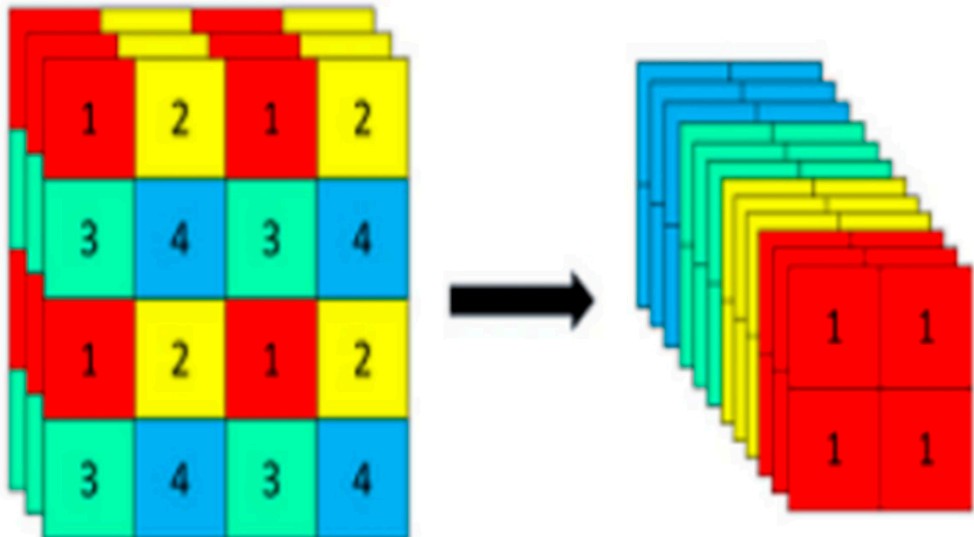

**Figure 2.** Focus structure. The red color represents the first three channels, the yellow and green colors represent the middle six channels, and the blue color represents the last three channels. Numbers represent sequence of the colors.

Neck: The neck network is mainly used to continue using the features extracted by the backbone network, and the feature pyramid type structure of FPN+PAN is used to process the feature maps extracted at different stages [33,34], so that the feature information can be passed under different scales of the object, as well as to solve the multi-scale challenges.

Head: After the neck network passes the fused feature map, the predictor head is responsible for predicting the features of the image, generating bounding boxes and predicting the category. The prediction head will use three grids of different scale sizes to detect small, medium, and large objects in the image, respectively.

The classroom behavior detection process based on the improved YOLOv5 model is shown in Figure 3. We use a BiFPN structure [35] to enhance the extraction of multi-scale features by modifying the feature pyramid structure in the neck network. A new link is established between the Concat layers and we perform adaptive multi-scale fusion on the three-layer CSP2_1 structure to improve the recognition accuracy using multi-scale small object fine-grained features [36]. The CBAM convolutional attention module [37] is further introduced after the multi-scale fusion weighting to enhance the saliency of the behavioral objects. The DIoU component with penalty terms is used to improve the non-maximum suppression and characterize the regression of object box in terms of centroid distance and overlap rate [38]. The following three aspects are introduced in detail from the feature pyramid structure, attention mechanism, and non-maximum suppression.

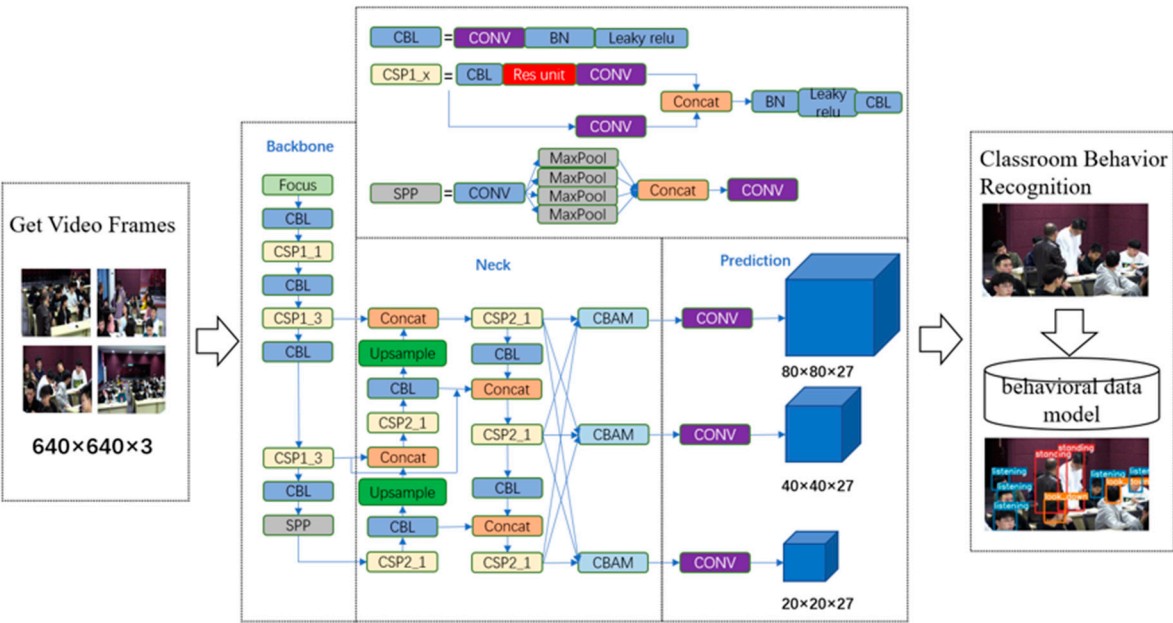

**Figure 3.** The network structure of the algorithm in this paper.

### 3.2. Feature Pyramid Structure

Different camera angles in the classrooms and the distances between students and the camera can lead to different size of students in the top and bottom rows of the classroom in the same image frame, so how to represent and deal with multi-scale features effectively is a difficult problem in classroom behavior detection. Figure 4a Feature Pyramid Network (FPN) proposes a top-down approach that combines multi-scale features for prediction using composite feature layers with more semantic information. In line with this idea, Figure 4b PANet continues to add a bottom-up path aggregation network on top of FPN, taking into account both the semantic information of the top layer and the location information of the bottom layer. The YOLOv5 framework integrates multi-scale features with the FPN+PAN structure, but due to the different resolution of input features, the contribution of the FPN+PAN structure to the fusion output features is often uneven and the features between different scales cannot be fully exploited. Therefore, a simple and efficient weighted bidirectional feature pyramid network (BiFPN) has been introduced to the neck of the YOLOv5 framework to improve the detection accuracy. The fusion steps are as follows:

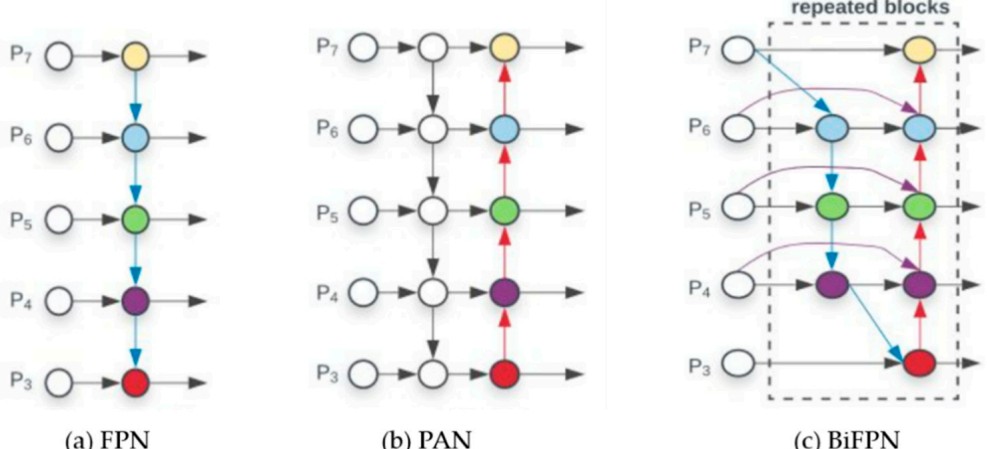

**Figure 4.** FPN, PAN, and BiFPN structures.

1. First remove those nodes that have only one input edge, because if a node has only one input edge without feature fusion, its contribution to a feature network that incorporates different feature information is usually weak. Therefore, BiFPN eliminates the intermediate nodes of P3 and P7 in PANet, which leads to a simplified bidirectional network. 2. Add a jump connection between input nodes to output nodes at the same scale because they are on the same layer and can incorporate more features and enhance feature representation without adding too much computational overhead. 3. Combine each bidirectional (top-down and bottom-up) path considered as a feature network layer so that it can be repeatedly stacked to achieve higher level feature fusion. The structure of the BIFPN in this paper is shown in Figure 5.

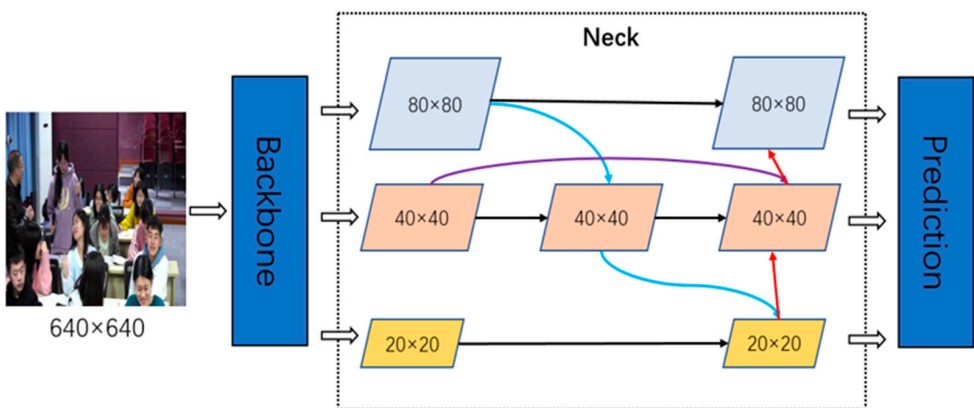

**Figure 5.** BiFPN feature fusion structure in this paper.

BiFPN uses the fast normalization method for weighted feature fusion, in which the fast normalization directly divides the weight by the weighted sum of all values for normalization, and the value of each normalization weight is also between 0 and 1, in order to increase the speed of calculation, the formula is as follows:

$$O = \sum_i \frac{\omega_i}{\in + \sum_j \omega_j} \cdot I_i \tag{1}$$

In the formula, $\omega_i$ represents the learnable weight, which is obtained by the network training, and $I_i$ represents the input feature. The weights $\omega_i \geq 0$ are ensured by using a ReLu activation function after each $\omega_i$, and the value of the output weights is controlled between 0 and 1 by regularization. Finally, BiFPN integrates bidirectional cross-scale connection and fast normalization fusion. Taking node P6 in Figure 6 as an example, the two feature fusion processes formed are as follows:

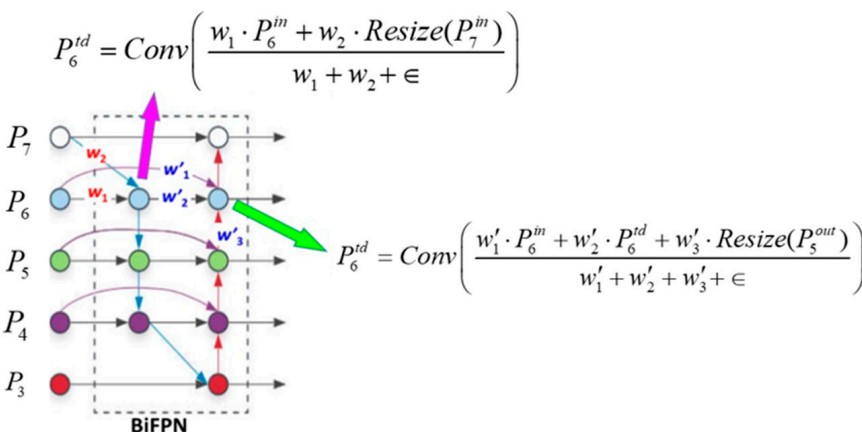

**Figure 6.** BiFPN feature fusion process.

In the formula, $P_6^{td}$ represents the top-down intermediate feature layer, $P_6^{out}$ represents the bottom-up intermediate special layer, Conv represents the convolution operation, and Resize represents the up-sampling or down-sampling operation. Based on the above advantages, BiFPN is introduced to improve the feature pyramid structure, enhance multi-scale feature fusion, and improve model detection accuracy.

### 3.3. Convolutional Attention Module CBAM

Because there are walls, desks, chairs, windows, and other background environments in the real classroom environment, when facing a large area of untargeted regions in an image, the YOLOv5 model is improved by introducing an attention mechanism in order to make the model better focus on the information of students' regions. The attention mechanism selects a small amount of important information from a large amount of information and focuses on this important information, ignoring most of the unimportant information. The Convolutional Block Attention Module (CBAM) used in this paper is a hybrid attention mechanism model that combines spatial and spatial channels; compared with SENAT [39], which only focuses on the channel itself, CBAM integrates both channel and spatial attention mapping processes and can retain more feature information. The structure of CBAM is shown in Figure 7.

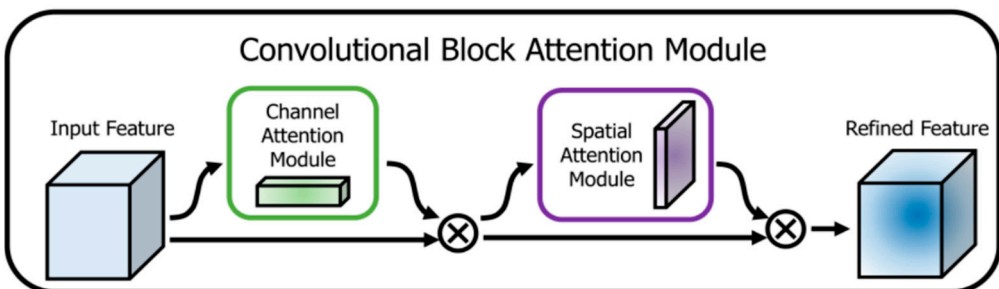

**Figure 7.** CBAM attention mechanism structure.

The above figure shows the overall flow structure of the CBAM module, which consists of two separate submodules, the channel attention module and the spatial attention module. The input features first pass through a channel attention module, which attaches the corresponding attention weights to each feature channel, and after getting the weighted results, they then pass through a spatial attention module, which applies the corresponding attention weights to different spatial locations of the feature map, and then weighted to obtain the output result of the convolutional layer.

Each channel of features represents a specific detector, so it makes sense for channel attention to focus on the features. The channel attention module compresses the feature map in the spatial dimension to obtain a one-dimensional vector and then operates. The channel attention module shown in Figure 8 first performs global max pooling and global average pooling on the input feature map in the width and height dimensions, respectively, in order to aggregate the spatial information of the feature map. The result is then passed into the multilayer perceptron for processing using the shared fully connected layer [40]. Finally, through the sigmoid activation operation, the final channel attention feature map $M_c(F)$ is generated. The result of this feature map, after weighting with the input feature map, is used as the input feature for the spatial attention module.

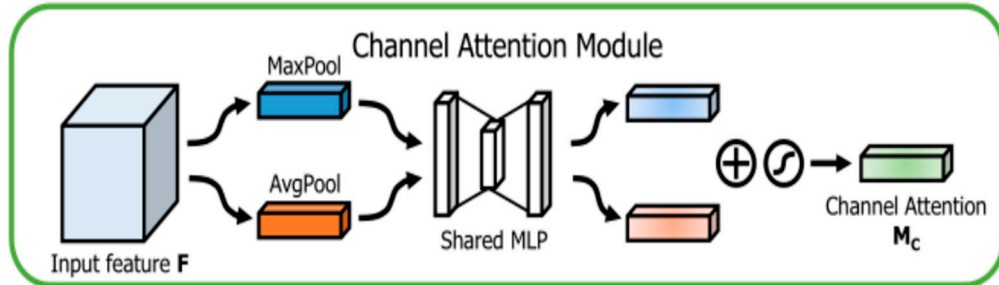

**Figure 8.** Channel attention mechanism.

The mathematical expression of the channel attention feature map $M_c(F)$ is:

$$M_c(F) = \sigma(MLP(AvgPool(F)) + MLP(MaxPool(F)))\tag{2}$$

where $\sigma$ is the sigmoid activation function, *MLP* is the connection weight operation between layers, and *AvgPool* and *MaxPool* represent the maximum pooling and average pooling operations, respectively.

After the channel attention module, we introduce the spatial attention module shown in Figure 9 to focus on where features make sense. Similar to channel attention, the spatial attention mechanism compresses channels, and performs mean pooling and max pooling in the channel dimension, respectively. First, the input feature layer takes the maximum value and average value on the channel of each feature point. After that, the two results are stacked, and the number of channels is adjusted by convolution with a channel number of 1. Finally, the spatial attention map $M_s(F)$ is generated by the sigmoid function, and its mathematical expression is:

$$M_s(F) = \sigma(f^{7\times7}([AvgPool(F); MaxPool(F)]))\tag{3}$$

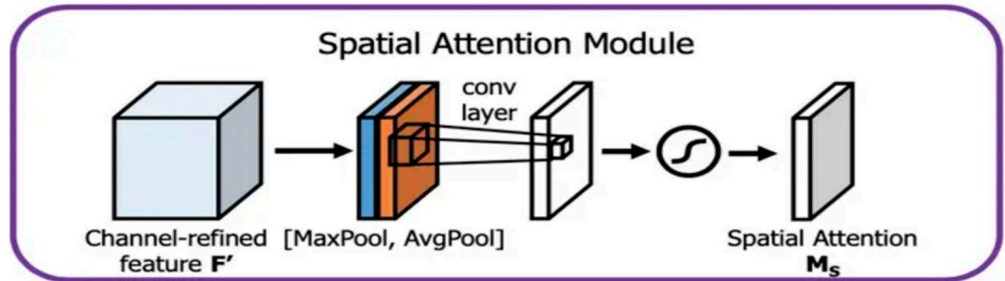

**Figure 9.** Spatial attention mechanism.

In the formula, $7 \times 7$ represents the size of the convolution kernel, that is, a $7 \times 7$ convolution operation is performed on the feature map. Experience shows that the convolution kernel of $7 \times 7$ is better than the convolution kernel of $3 \times 3$. This paper adds the CBAM attention mechanism between the neck network and the prediction network.

As can be seen in Figure 10, after adding the attention mechanism, the algorithm proposed in this paper focuses more on the student targets in the classroom, ignoring the irrelevant factors in the cluttered background of the classroom. Using saliency of the objects is beneficial by improving the learning of student target features and improving the detection accuracy.

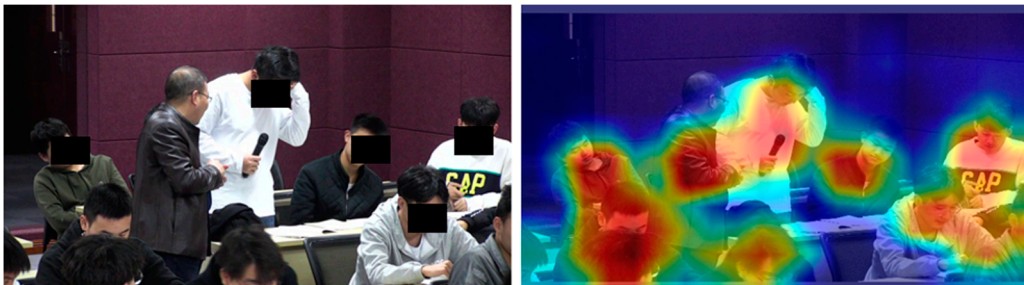

**Figure 10.** Feature visualization after adding attention mechanism.

*3.4. Non-Maximum Suppression Improvement*

In the prediction stage, the object detection algorithm usually uses non-maximum suppression to eliminate the redundant detection frames. The idea is to traverse all detection frames and retain the highest-scoring prediction frame, calculate the intersection ratio IoU between the remaining frames and the highest-scoring prediction frame, and reject the detection frame if the IoU is greater than the set threshold. In most cases, the NMS method is effective, but in the case of dense target scene, due to the large number of detected objects and serious occlusion, the objects to be detected are too close to each other, and the overlapping area of detection frames of adjacent objects is too large, which easily leads to the NMS incorrectly rejecting the detection frames of a certain object. In the experiments of this paper, the classrooms are densely packed with students and heavily occluded, and the traditional NMS is not applicable to the dataset of this paper, so DIoU is introduced in this paper to improve the post-processing NMS process.

The blue box in Figure 11 is the real frame, and its center point is marked as $b^{gt}$, the green box is the predicted frame, and its center point is marked as $b$, and the outer frame is the smallest box that wraps both the real frame and the predicted frame. Denoted as $C$, $c$ is the length of the diagonal of the outer frame, $d$ is the length of the center of the real frame and the center of the predicted frame, and the gray shaded part is the intersection of the real frame and the predicted frame.

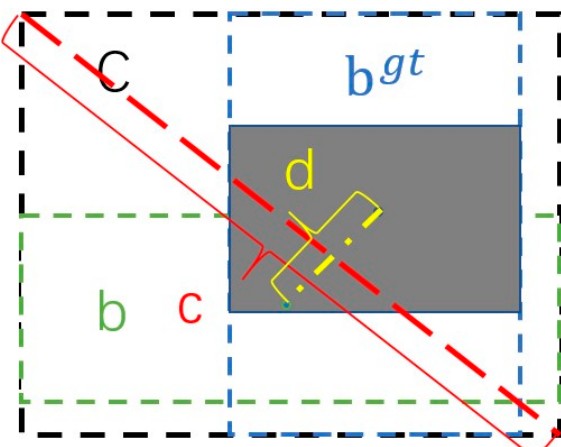

**Figure 11.** Example of detection frame.

Compared with IoU, DIoU takes into account the distance, overlap rate, and scale between the object and the anchor, making the object box regression more stable, as shown in formula:

$$\text{DIoU} = \text{IoU} - \frac{\rho^2(b, b^{gt})}{c^2} \tag{4}$$

In the formula, $\rho^2(\cdot)$ represents the Euclidean distance, and the definitions of $b$, $b^{gt}$, and $c$ are the same as in Figure 9.

The definition of DIoU is as formula:

$$s_i = \left\{ \begin{array}{l} s_i, \mathrm{DIoU}(M, B_i) < \varepsilon \\ 0, \mathrm{DIoU}(M, B_i) \geq \varepsilon \end{array} \right\} \tag{5}$$

In the formula, $s_i$ represents the classification score, $M$ represents the prediction frame with the highest score, $B_i$ represents the prediction frame to be eliminated, and $\varepsilon$ represents the artificially set IoU threshold. Compared with IoU, the calculation of DIoU takes into account the information of the center points of the two frames, which helps to solve the occlusion problem caused by the object distance being too close. Therefore, the NMS effect judged by DIoU is more realistic and the effect is better.

## 4. Experimental Results and Analysis

### 4.1. Dataset Production

Existing object detection public datasets, such as COCO and PASCAL VOC, are used to detect specific categories, and there is no public data information for classroom students' behavioral states. Therefore, this paper manually constructed a classroom behavior detection dataset, based on real classroom videos, which contains four types of behaviors: listening, looking down, lying down, and standing. As shown in Figure 12, students who looked at the blackboard or read books carefully were labeled as listening state. Students who lowered their heads or played with cellphones were labeled as looking down state. Students who lay on their sides or buried their heads were labeled as lying down state. Students who stood up and answered questions were labeled as standing state. Since the head features of listening and looking down states were more obvious, only the head region was marked in this paper, while lying down and standing need to be identified by body posture, so the whole-body region was marked.

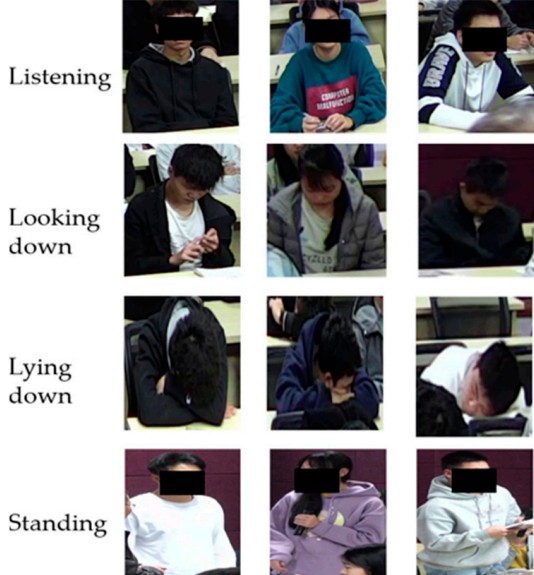

**Figure 12.** Some samples of this dataset.

Dataset images were captured at frame intervals from recorded class videos. In order to ensure good generalization performance and robustness, each image contains multiple student objects and multiple action poses. The images were screened out with insignificant actions and small changes before and after, and the LabelImg tool was used to manually label each image. The xml file generated by each image includes the label category (listening, looking down, lying down, standing) and the coordinate information of the real frame. A total of 10,826 labels were marked, and the training set and test set were divided by 8:2. The number of labels for each label is shown in Table 1.

**Table 1.** The number of labels in the student behavior dataset.

| Category | Training Set | Test Set | Total |
|---|---|---|---|
| Listening | 3105 | 776 | 3881 |
| Looking down | 2742 | 685 | 3427 |
| Lying down | 1843 | 460 | 2303 |
| Standing | 972 | 243 | 1215 |
| Total | 8662 | 2164 | 10,826 |

Unbalanced training data will cause the network model to pay more attention to a large number of objects during the training process, and it is easy to cause the model to overfit. In the dataset of this paper, the number of standing samples is relatively small, and in real life, sample collection is not easy. Therefore, we used data augmentation to expand the number of samples of standing poses to improve network performance. In this paper, a selection of images was intercepted of the student objects in the standing posture and data enhancement was carried out. Horizontal flip, brightness enhancement, and Gaussian noise were added to the cropped image, and the enhancement effect is shown in Figure 13.

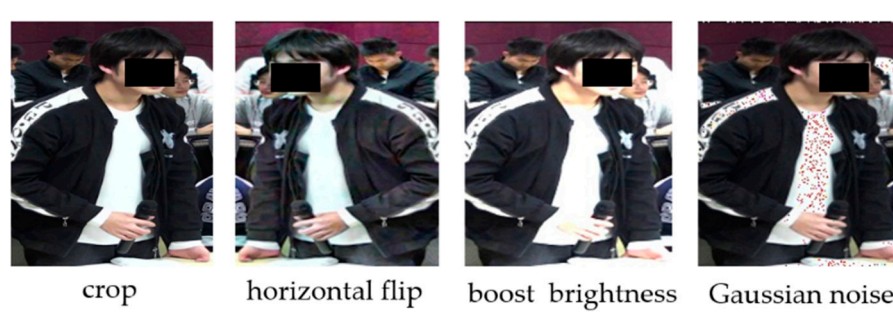

crop          horizontal flip          boost brightness          Gaussian noise

**Figure 13.** Data enhancement effect.

### 4.2. Experimental Settings and Model Training

In this paper, the experimental environment was conducted on Ubuntu 18.04 operating system, the CPU configuration was Intel(R) Xeon(R) Sliver 4110 CPU@2.10HZ, the graphics card GeForce GTX 1080ti was selected for computing, the video memory size was 11 GB, and the runtime environment was built using the deep learning framework Pytorch version 1.7.0, and the Python environment was 3.6.13.

The image input was set to $640 \times 640$ size, the momentum size parameter was set to 0.937, and the initial learning rate was set to 0.01. The warm-up method was used to warm up the learning rate and slow down the phenomenon of overfitting the model to small batch data at the early stage of training. The learning rate was updated by one-dimensional linear interpolation in the warm-up phase, and the learning rate was adjusted by the cosine annealing strategy after the warm-up. The total number of epochs was 100, the epoch batch size was set to 16, the SGD optimizer was used, and the mosaic data enhancement strategy was used for the first 90 epochs and turned off for the last 10 epochs.

### 4.3. Evaluation Indicators

In order to evaluate the superiority of the algorithm objectively, the evaluation indicators use the precision rate, the recall rate, the mean average precision mAP, and the number of frames per second to detect the pictures to evaluate the performance of the algorithm in this paper. The calculation formula for each indicator is as follows:

$$\text{Precision} = \frac{\text{TP}}{\text{TP} + \text{FP}} \tag{6}$$

$$\text{Recall} = \frac{\text{TP}}{\text{TP} + \text{FN}} \tag{7}$$

$$\text{mAP} = \frac{\sum_{j=0}^{n} AP(j)}{n} \tag{8}$$

$$\text{FPS} = \frac{\text{Figure Numbers}}{\text{Total Time}} \tag{9}$$

Among them, TP, FP, and FN represent the number of correct detections, the number of false detections, and the number of missed detections, respectively, $AP(j)$ is the average precision of the $j$th type of defects, Figure Numbers represents the total number of detected pictures, and Total Time represents the total detection time.

*4.4. Experimental Results*

The comparison results of the average precision value curve of the improved network model in this paper and the YOLOv5 model are as follows:

In Figure 14, mAP@0.5 represents the average precision when the IoU value is 0.5, and mAP@0.5:0.95 represents the average mAP when the IoU value is 0.5 to 0.95. As can be seen from the figure, the average accuracy of this algorithm is significantly improved compared to YOLOv5. When the algorithm in this paper iterates to the 40th round, mAP@0.5 rises to about 0.8, and finally increases steadily to about 0.898, while the YOLOv5 algorithm iterates to the 60th round. The mAP@0.5 only rose to 0.8 in the round, and finally stabilizes at around 0.842.

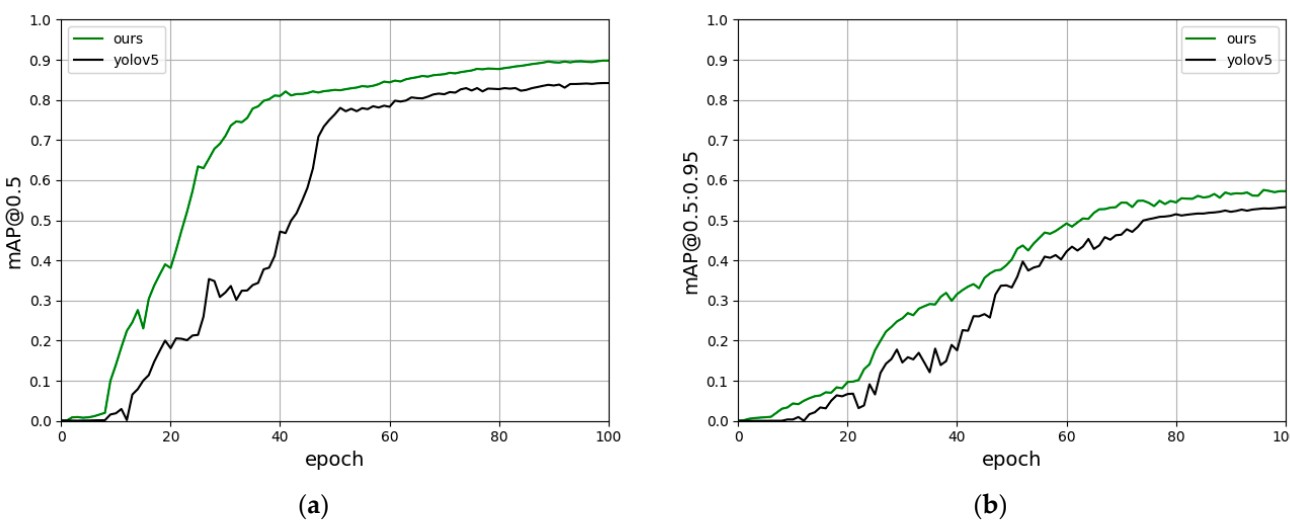

(**a**)                    (**b**)

**Figure 14.** Comparison of the average precision value curve of the algorithm: (**a**) mAP@0.5; (**b**) mAP@0.5:0.95.

As can be seen in Table 2, the algorithm performed well in the classroom behavior detection dataset, with mAP@0.5 reaching 89.8 and the average recall rate reaching 90.4. The detection accuracy of standing position is higher because the standing position occupied a more obvious image area, so the detection accuracy of standing position reached 93.3. The detection accuracy of listening and lying down behaviors is slightly lower than other behaviors because the area occupied by head is less.

**Table 2.** Average precision and recall for each behavior.

| Category | mAP@0.5 (%) | Recall (%) |
|---|---|---|
| Listening | 88.7 | 89.4 |
| Looking down | 86.9 | 85.8 |
| Lying down | 90.4 | 91.7 |
| Standing | 93.3 | 94.8 |
| Average for all categories | 89.8 | 90.4 |

As can be seen from Table 3, the detection accuracy of the algorithm in this paper is improved by 5.6% compared with the original YOLOv5 algorithm. In order to get a more intuitive feeling of the detection effect of the improved algorithm in this paper and YOLOv5, the images under the dense scene of classroom and the obscured scene were selected for the detection comparison, and the results are shown in Figures 15 and 16.

**Table 3.** Comparison before and after adding different modules.

| Model | CBAM | BiFPN | DIoU | Precision/% | Recall/% | mAP@0.5/% |
|---|---|---|---|---|---|---|
| YOLOv5 | × | × | × | 83.7 | 85.3 | 84.2 |
| +CBAM | √ | × | × | 87.1 | 90.1 | 88.7 |
| +BiFPN | × | √ | × | 86.5 | 89.3 | 87.4 |
| +DIoU | × | × | √ | 85.4 | 87.6 | 85.9 |
| ours | √ | √ | √ | 88.2 | 90.4 | 89.8 |

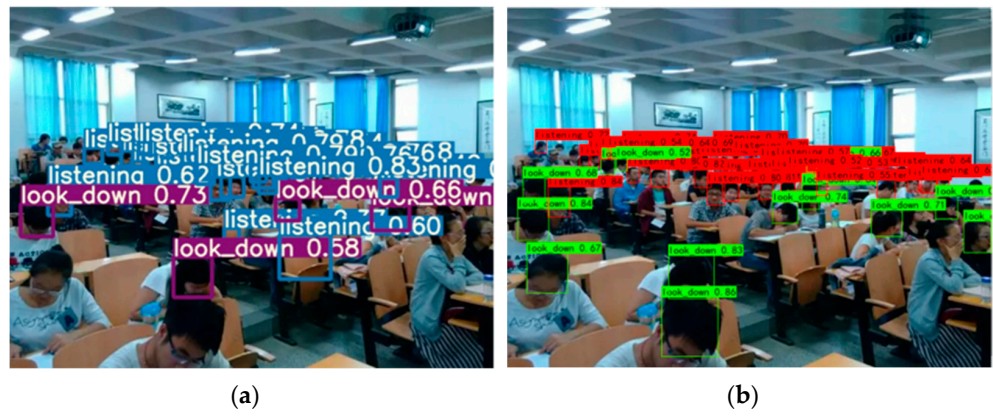

(a)  (b)

**Figure 15.** Comparison of dense scene detection effects. (**a**) YOLOv5. (**b**) Ours.

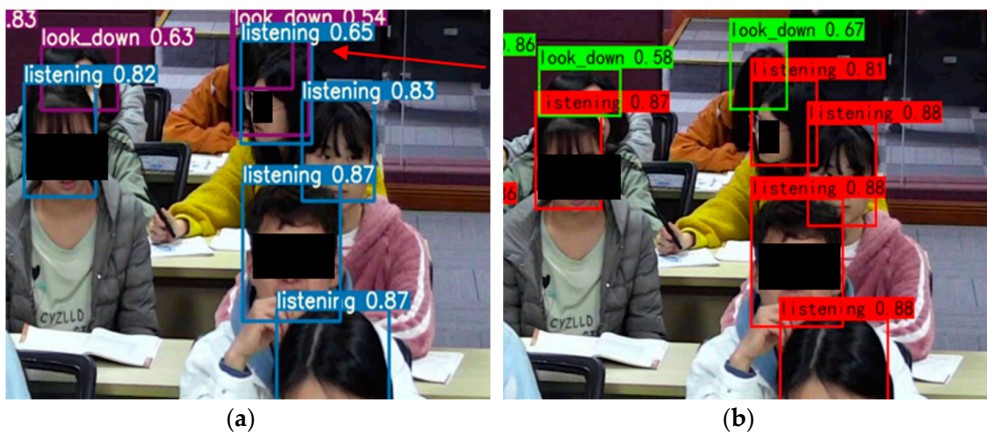

(a)  (b)

**Figure 16.** Comparison of occlusion scene detection effects. (**a**) YOLOv5. (**b**) Ours.

From the test results, it was found that the YOLOv5 model will have some missed detections in the classroom scene with intensive student targets, and the detection effect of the small target students in the back row is not satisfactory. The improved algorithm in this paper can not only identify the missing object of YOLOv5, but also most object recognition is more accurate than the YOLOv5 model. This is due to the introduction of the CBAM attention mechanism and the improvement of the BiFPN feature pyramid, which allow our model to better handle multi-scale features and capture more high-level semantic information, and then focus more attention on the object features of the image to be detected, and the accuracy of the final recognition increases.

For images in occluded scenes, when there are multiple student targets interfering with each other, the YOLOv5 model does not suppress redundant detection frames, and sets two detection frames for the occluded targets. Based on the introduction of DIoU to improve non-maximum suppression, the algorithm in this paper calculates the distance between the center points of adjacent target detection frames, so that the prediction frame can predict targets more accurately and eliminate redundant detection frames.

In order to evaluate the advantages of the algorithm objectively, the algorithm in this paper is compared with the mainstream object detection algorithms. To ensure fairness, all algorithms in the experiment use the same training parameters and the same training samples, and the experimental results are shown in Table 4.

**Table 4.** Comparison and detection results of each algorithm.

| Algorithm | mAP@0.5/% | FPS |
| --- | --- | --- |
| Faster R-CNN | 81.5 | 7.1 |
| SSD | 73.2 | 32.5 |
| YOLOv5 | 84.2 | 41.6 |
| Ours | 89.8 | 33.8 |

It can be seen from the comparison results of the algorithms in Table 4 that the average accuracy of the improved algorithm in this paper is higher than that of the other three algorithms. Because of the improved feature pyramid structure and the introduction of the attention mechanism, the detection speed of the algorithm in this paper is slightly lower than the YOLOv5 model, but it can still detect 33 frames of images per second, achieving the goal of detecting student behavior in real time.

In order to verify the robustness of the algorithm, two kinds of Gaussian noises with different variance were added to the images of long, medium, and short-range camera scenes. The experimental results are shown in Table 5 and Figure 17.

**Table 5.** Detection results after adding different variance noises.

| Gaussian Noise | mAP@0.5/% | Recall/% |
| --- | --- | --- |
| variance 0.01 | 89.3 | 89.6 |
| variance 0.05 | 88.5 | 88.7 |

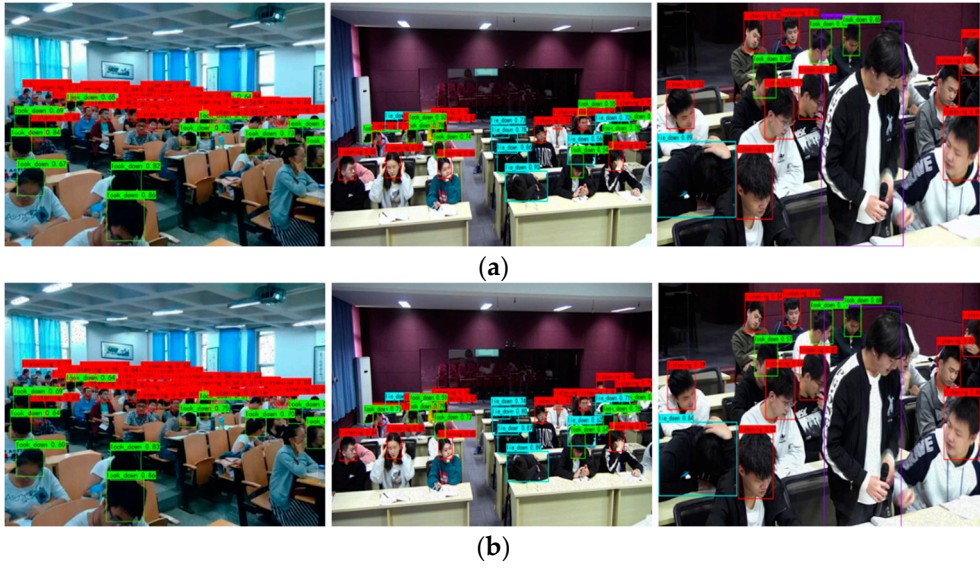

**Figure 17.** Detection effect after adding noise. (**a**) variance 0.01. (**b**) variance 0.05.

As can be seen from Figure 17, the algorithm in this paper can still accurately detect a variety of classroom behaviors after adding Gaussian noise. As can be seen from Table 5, mAP@0.5 and Recall were still high even though the accuracy of the algorithm decreases slightly with the addition of noise, proving the robustness of our algorithm.

## 5. Conclusions and Discussions

This paper detects students' classroom behavior based on the improved YOLOv5 algorithm and transforms the problem of classroom behavior detection into a fine-grained classification problem of behavior images. By introducing the weighted bidirectional feature pyramid network BiFPN, the multi-scale features of the object are more effectively expressed, mining the fine-grained characteristics of object behavior, reducing the number of missed detections of small target students in the back row of the classroom; the attention mechanism CBAM is integrated into the YOLOv5 algorithm to enhance the saliency of the object area in the complex background, effectively improving the accuracy of the detector; and using DIoU to improve the original non-maximum suppression, further improving the accuracy of the detector, while improving the problem of false detection caused by the problem of classroom occlusion.

The experimental results show that the improved algorithm in this paper can accurately detect different classroom behaviors and has higher accuracy than the YOLOv5 algorithm in dense classroom scenes and occlusion scenes, and has better accuracy and robustness. The mAP values of the proposed algorithm are bigger than those values of Faster R-CNN, SSD and the original YOLOv5 algorithm, indicating that the proposed algorithm has a better detection accuracy. The results presented in Figures 15 and 16 also show that the algorithm proposed in this study can be used for detection in dense classroom scenes and occlusion scenes. Moreover, the results show that the proposed algorithm achieves an inference speed of 33 frames per second under GeForce GTX 1080ti, which meets the requirements of real-time video detection.

Although the results are encouraging, the work is limited due to the following reasons. First, students' behaviors often occur coherently and video datasets in real classrooms are difficult to obtain. Second, it is difficult to consider the temporal contextual information of behavior occurrence to increase the accuracy. We did not consider the temporal features before and after each behavior occurrence but focused on the behavior classification problem on single-frame images. Unlike running and jumping that require multiple frames of information, the behaviors performed by students in the classroom often tend to be static and can be judged by single-frame images.

The findings of this study have the following potential applications in the classroom scenarios. To monitor students' behavioral status, it is very common to arrange cameras in front of the classroom and connect it to our back-end server. Deploying the algorithm proposed in this study, the way in which students behave can be presented in a timely manner on both the teacher and administration sides. The status of students can be evaluated automatically by classifying their behaviors into positive behaviors (e.g., listening and standing), neutral behaviors (e.g., looking down), and negative behaviors (e.g., lying down). The educational system calculates the proportion of positive, neutral, and negative behaviors by counting the recognized behaviors. If the proportion of negative behaviors exceeds a certain threshold, feedback can be provided to the teacher and suggest he/she improves the teaching strategy of the class. The educational system can also count the number of times that each student's behavior occurs and the period during which positive or negative behavior lasts. In this way, teachers can get more insights on the teaching method that is more appropriate for them at different teaching stages.

Despite the performance of the algorithm being improved, the accuracy of student behavior recognition for complex features or state combinations in practical application environments needs to be further investigated. Further work to be undertaken includes the use of contextual information of behaviors for more accurate behavior classification and the accuracy improvement for small target students. Doing this, it is necessary to

expand the dataset by adding more behavior types and improving the generalizability of the algorithm in diverse classroom scenarios. The coordinate information of the student's position and face features can be utilized to achieve one-to-one correspondence between student identification and their behavior status in the smart classroom.

**Author Contributions:** Conceptualization, L.T. and Y.Y.; methodology, Y.Y.; software, L.T.; validation, L.T., T.X. and Y.Y.; formal analysis, H.W.; investigation, L.T.; resources, L.T.; data curation, H.W.; writing—original draft preparation, L.T.; writing—review and editing, T.X.; visualization, L.T.; supervision, Y.Y.; project administration, T.X.; funding acquisition, Y.Y. All authors have read and agreed to the published version of the manuscript.

**Funding:** This research was funded by Chongqing Educational Planning Project under grant number 2017-GX-268, Chongqing Education Commission Science and Technology Research Project under grant number KJQN201800534, and Teaching Reform Research Project of Chongqing Higher Education under grant number 213082.

**Informed Consent Statement:** Informed consent was obtained from all subjects involved in the study.

**Data Availability Statement:** The data presented in this study are available on request from the corresponding author. The data are not publicly available due to privacy restrictions.

**Conflicts of Interest:** The authors declare no conflict of interest.

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
