# Peer review of "Classroom Behavior Detection Based on Improved YOLOv5 Algorithm Combining Multi-Scale Feature Fusion and Attention Mechanism"

_applsci, doi:10.3390/app12136790_

Round 1
Reviewer 1 Report
The description of the methods does not have to be that exact. It can be shortened.
In this article, the authors used classroom behavior detection which included four types of behavior: listening, look_down, lie_down and standing. In the future, the authors may expand the types of behavior.
Reviewer 2 Report
Authors in paper correctly suggests algorithm based on the improved YOLOv5 model is proposed. Authors by comparing the effects of student behavior detection in different scenarios, was created the improved algorithm has an average accuracy of 89.8% and a recall of 90.4% for classroom student behavior detection. These results should be considered satisfactory.
My comments to the article are as follows:
- I propose to extend the background in the field of artificial intelligence methods, by citing, for example: The use of multilayer ConvNets for the purposes of motor imagery classification, Automation 2021: Recent achievements in automation, robotics and measurement techniques, Series: Advances in Intelligent Systems and Computing, Springer from the year 2021. This will expand the background of the article regarding possible implementations of the Convolutional Neural Network. In addition, you will update the bibliography with one of the newer publications.
- For ease of interpretation of the content in the article, I propose to separate the Results section from the Analysis section.
- In my opinion, Conclusions should be extended to include more detailed plans for the future in the scope of the conducted research.
Reviewer 3 Report
Journal name: Applied Sciences (ISSN 2076-3417)
Title: Classroom behavior detection based on Improved YOLOv5 algorithm combining multi-scale feature fusion and attention Mechanism
Manuscript ID: applsci-1774455
Section: Computing and Artificial Intelligence
Special Issue: Technologies and Environments of Intelligent Education
Abstract
Abstract: To detect students' behaviors in the classroom to provide a reference for assessing the effectiveness of classroom teaching. A classroom behavior detection algorithm based on the improved YOLOv5 model is proposed. First, the FPN+PAN structure in the neck network of the original YOLOv5 model is improved to a weighted bidirectional feature pyramid network BiFPN to fully fuse the features of different scales of the target. Second, adding a CBAM spatial and channel convolutional attention mechanism between the neck network and the prediction network makes the model pay more attention to the target information to improve the detection accuracy. Finally, replacing the original non-maximal suppression with DIoU_NMS improves the recognition of occluded objects. To address the problem of sparse classroom behavior dataset and no publicly available large dataset, we built our own classroom behavior data sets, and used data augmentation algorithms to expand the data sets. The improved algorithm can accurately detect various student behaviors in this dataset, and the accuracy is higher than that of the YOLOv5 model. By comparing the effects of student behavior detection in different scenarios, the improved algorithm has an average accuracy of 89.8% and a recall of 90.4% for classroom student behavior detection, both of which are better than other detection algorithms.
Thank you for inviting me to review this work. First of all, I would like to thank and acknowledge the effort and work done by the authors on this study.
In order to follow and understand the comments made on your work, I inform you that I will respect the order and structure of your manuscript.
In the abstract, you must first state the meaning of the acronyms you use and then you can use them throughout the text. Please do this with the acronyms YOLOv5; FPN+PAN; BiFPN...CBAM....
The abstract should specify the sample selection procedure and the characteristics of the sample in brief. It is intended to detect the behaviors of the students in the classroom... the type of behaviors to be observed to validate the algorithm's determination capacity is not specified.
In the introduction of the paper, the authors state: adds temporal dimension to the 2D CNN [5], and learns both spatial and temporal information. Hara et al. [6] proposed ResNet6based on 3D network. On this basis, a pseudo 3D… ... please remove the names of the authors of the text and cite according to the journal's style guidelines.
List authors such as the one you cite in section 2. Related Work: …de Girshick11 et al. [11] proposed the first two-stage target detection algorithm R-CNN. First, the region proposal network was used to extract candidate frames from the image… revise the entire document, and delete the names of the authors.
In relation to the theoretical framework, I believe that the authors should address a key aspect mentioned throughout the study. If motivation is studied in relation to the detection of certain movements, posture, etc., aspects that deepen this relationship should be addressed.
A sample section should be added with a description of the observation scenario, the characteristics of the observed subjects...
The authors do not address the effect of aspects and elements of the context such as the type of activity they perform, the times when they perform them, previous levels of motivation, etc... How do they explain not having considered these aspects?
The study is experimental in nature, didn't you perform a control of the behaviors in another control group?
The authors should add a section on the implications of the results of this study. How can they improve the conditions for learning in the classroom? How could teachers use them?
Authors should also add a study limitations section.
Ethical aspects are not mentioned in this study. Since this is an experimental study, the authors should explain all the ethical aspects of the experiment. Was the study approved by an ethics committee? If so, indicate the name and date of approval. How was the problem of handling sensitive information, such as the images that appear in this work, solved?
Overall, the topic is of interest and the contributions are interesting, however, the authors should make the suggested changes and resubmit the paper. Please review the paper carefully to include the changes appropriately.
Please note that your paper requires improvements in the theoretical framework, mainly in the methodology section. The changes should be a substantial improvement to the paper.
The document must be reviewed by a native English speaker to correct grammar, expression, and vocabulary.
Please take the time to improve your papers.
Best regards
Round 2
Reviewer 3 Report
Thank you for addressing the comments and improving the work. The authors have corrected the errors and suggested changes. The quality of the work has greatly improved.
I only suggest to the authors one last change they have not addressed correctly.
In the section addressing ethical issues, the authors should indicate that: consent was obtained from the participants for participation in the study and for using images.
